# HENP: Dynamic Pruning via Neuron Entropy

## Abstract

We introduce a novel framework for analyzing neural networks based on the concepts of *dynamic* and *static* neurons, which describe the stability of neuron activation under specific inputs. From these concepts, we propose *neuron entropy* as a metric to quantify network expressiveness. Our analysis reveals that better generalization correlates with diverse activation patterns and higher neuron entropy. Building on this, we propose our HENP method, a dynamic pruning technique that regulates dying neurons and sparsifies the network during training. Experimental results demonstrate that our HENP improves both network sparsity and performance, offering a new approach to efficient neural network optimization.

## 1 Introduction

Deep neural networks equipped with piecewise linear activation functions, such as the Rectified Linear Unit (ReLU), have gained widespread popularity due to their remarkable performance across various domains and computational efficiency Zhou (2021); Jiang (2018). However, understanding the underlying mechanics of these networks remains a significant challenge. In this paper, we propose a comprehensive framework for analyzing the structure and behavior of deep neural networks, specifically focusing on the partitioning of the input space of a network $\mathcal{N}$, defined on $\mathbb{R}^n$ with a piecewise linear activation function $\pi$. This partitioning results in multiple linear zones, where the mapping within each zone is linear, collectively enabling the approximation of complex, non-linear functions.

Recent studies have established that the number of linear zones serves as a quantitative measure of a model's complexity Cohan (2022). However, as the depth and width of a network increase, the significance of analyzing a single linear zone diminishes. To provide a more nuanced evaluation of large-scale models, our framework investigates neuron activation across multiple linear zones, capturing not only the linear transformations within individual zones but also the intricate transitions between them, which are crucial for enhancing the performance of deep neural networks.

By analyzing neuron behavior across different activation zones, we gain insights into the representational capacity and stability of these models, addressing the challenges posed by the increasing complexity of modern architectures. Our contributions include a detailed examination of the fundamental structures and formalisms of feedforward neural networks, as outlined in Section 3. In Section 4, we explore the global stability of neuron activation zones, revealing that dynamic neurons significantly enhance the model's ability to capture complex functions. Conversely, a high proportion of static neurons can lead to linear approximations, undermining representational capabilities. To quantify neuron uncertainty, we introduce a novel neuron entropy metric, demonstrating its correlation with representation capacity in deeper layers.

Furthermore, we address the issue of dying neurons Ziping (2023); Lu (2020), where activations remain unchanged regardless of input. To mitigate this, we propose a Hybrid Entropy-Norm Pruning (HENP) technique, enabling effective network pruning without additional training or accuracy loss by eliminating these dying neurons. In Section 5, we detail the integration of norms and entropy in HENP and present experimental results illustrating its effectiveness, including the computation of neuron entropy on the test set and the application of a masking strategy on neurons with the lowest entropy.

## 2 RELATED WORKS

### 2.1 ON NEURAL NETWORKS ANALYSIS

To understand the remarkable performance of neural networks, early research addressed fundamental challenges like vanishing and exploding gradients Bengio (1994); Hochreiter & Schmidhuber (1997), prediction instability Szegedy (2013), and constraints on model capacity Montúfar (2014). These challenges spurred extensive efforts to analyze the internal workings of neural networks Goodfellow (2017); Xavier & Bengio (2010), utilizing methods such as feature attribution Samek (2017); Montavon (2018); Ancona (2018) and complexity analysis Raghu (2017). Feature attribution techniques, such as Integrated Gradients Sundararajan (2017), enhance interpretability, which is crucial for transparency and trust in neural networks Doshi-Velez & Kim (2017). In parallel, theoretical research has advanced, particularly regarding over-parameterization and model complexity trade-offs Bubeck (2020), as well as the linear zones of ReLU networks, which relate to expressivity and approximation capacity Zhang (2021); Pascanu (2013). There is also a growing emphasis on interpretable models, with methods integrating decision trees Murdoch (2019); Letham (2015) or prototype-based classifiers like ProtoPNet Chen (2019), offering human-like recognition and transparency Caruana (2015). Further studies have examined the piecewise linear properties of networks Arora (2018), quantifying their complexity Hanin & David (2019), which informs generalization capabilities Goodfellow (2015).

### 2.2 TOWARDS NETWORK COMPRESSION

Model compression is a crucial research area in deep learning, particularly for deploying large models on resource-constrained devices. The goal is to reduce computational and memory costs while maintaining predictive accuracy. Techniques include *pruning*, which removes redundant weights or neurons, with structured pruning targeting filters or channels to enhance hardware compatibility Blalock (2020); Lemaire (2019). *Quantization* lowers the precision of weights and activations, typically from 32-bit to 8-bit, using post-training or quantization-aware methods to preserve accuracy Krishnamoorthi (2018); Jacob (2018); Gholami (2021). *Knowledge distillation* trains smaller models to replicate the output of larger ones, efficiently transferring knowledge without significant performance loss Hinton (2015); Tian (2022); Gou (2021). *Neural Architecture Search* (NAS) automates the design of efficient models, employing multi-objective optimization for accuracy and speed, while recent advancements like differentiable NAS reduce search costs, making it more feasible for hardware-constrained deployment Elsken (2019); Cai (2020); Radosavovic (2020); Xu (2020); Zela (2020). Together, these techniques drive significant improvements in compression, enabling more practical applications of deep learning Liu (2019); Han (2016); Cheng (2018).

### 2.3 UNDERSTANDING DYING NEURONS

The ReLU activation function can lead to neuronal death, where neurons stop contributing to the network due to weight updates. While ReLU avoids gradient vanishing issues common in sigmoid functions and helps prevent gradient saturation, its drawbacks include neuron inactivation when the learning rate is too high. Specifically, the update rule $w' = w - \eta \Delta w$, where $\eta$ is the learning rate and $\Delta w$ is the gradient, can result in $w'$ becoming negative if $\eta \Delta w$ exceeds $w$. This negativity causes inputs to be zeroed after passing through ReLU, and once ReLU outputs 0, its derivative is also 0, leading to permanent inactivation of the neuron as the gradient no longer updates the weights. Solutions such as resetting dead neurons or concatenating ReLU activations can mitigate this issue Whitaker & Whitley (2023); Utku (2020); Mike (2023); Joo Hyung (2023); Sokar (2023); Abbas (2023). Neurons that consistently receive negative inputs face a similar problem, as their weights and biases cannot be updated due to zero gradients. This results in neuron "death," where certain neurons remain inactive throughout training, never contributing to the learning process Ziping (2023); Lu (2020).

## 3 MATH FUNDAMENTALS OF NEURONS

In this section, we introduce the formalisms and describe the fundamental structure of a feedforward neural network for a classification task, which will serve as the basis for our analysis of activation

zones, zones, and flows in subsequent sections. (see Appendix A.1, A.2, A.3 and A.4). Due to space constraints, the detailed discussion of these concepts is provided in subsequent sections.

## 3.1 Consistency of Neuron States and Trajectories

The behavior of trajectories can be categorized based on the consistency of the neurons along them. We define dynamic and static neurons and trajectories as follows:

**Definition 1** (Dynamic and Static Neuron States). *A neuron $n_i$ is called a dynamic neuron if its activation state varies within a subspace $\mathcal{S} \subseteq \mathbb{R}^n$; otherwise, it is a static neuron.*

**Definition 2** (Dynamic and Static Activation Flows). *A flow $\tau(\mathbf{x})$ is static if all neurons along it are consistent within a zone $\mathcal{S} \subseteq \mathbb{R}^n$; otherwise, it is a dynamic flow.*

Static trajectories contribute consistent outputs within a zone, while dynamic trajectories introduce more nonlinearity and variability.

## 3.2 Dynamic Neuron Coverage

We explore the concept of dynamic neurons within a convex activation zone. The following lemma describes how the activation status of dynamic neurons can define a cover for the zone.

**Lemma 1** (Dynamic Neuron Coverage Lemma). *For any convex activation zone $\mathcal{C}$, let the set of dynamic neurons be denoted as $\mathcal{F}$. Then the convex activation zone $\mathcal{C}$ can be covered by a union of smaller zones defined by the activation status of the dynamic neurons. Specifically, the activation status of all static neurons remains the same throughout this zone, while the status of the dynamic neurons may vary across the different sub-zones.*

This lemma provides a method to approximate the coverage of a convex activation zone by focusing on the dynamic neurons, which introduces flexibility in the activation zones within the zone.

# 4 Dying Neurons and Model Expressivity

This section explores the expressive capacity of feedforward neural networks by analyzing the diversity of neuron activation zones. To evaluate the network's expressiveness, we introduce a metric called *neuron entropy*, which captures the *uncertainty* in neuron activation zones. Subsequently, we examine the relationship between this metric and both model performance and metrics proposed in previous research. Our findings indicate that in the deeper intermediate layers, a considerable number of neurons maintain identical activation regardless of the input, resulting in a reduced expressive capacity. (see Appendix A.5, A.6 and A.7). Due to space constraints, the detailed discussion of these concepts is provided in subsequent sections.

## 4.1 Neuron Entropy: A Key to Expressivity

**Definition 3** (Neuron Entropy). *Let $\mathcal{N}$ be a network defined by Definition A.1. The entropy of a neuron $(i, j)$ in $\mathcal{N}$ for a data distribution $\mathcal{D}$ is defined as the entropy of the distribution of the activation status for $(i, j)$ given input $(\mathbf{x}, y) \in \mathcal{D}$:*

$$\mathcal{E}_j^{(i)}(\mathcal{N}, \mathcal{D}) = -\sum_{k=0}^{q} p(\hat{a}_j^{(i)}(\mathbf{x}) = k) \log(p(\hat{a}_j^{(i)}(\mathbf{x}))). \tag{1}$$

The neuron entropy describes the uncertainty of the neuron activation zone. For instance, let $\mathcal{N}$ be a neural network with a ReLU activation function. If $\mathcal{E}_j^{(i)}(\mathcal{N}, \mathcal{D}) = 0$, this suggests that the activation zone of neuron $(i, j)$ is identical for any data $(\mathbf{x}, y) \sim \mathcal{D}$. This implies that neuron $(i, j)$ fails to provide non-linearity, behaving as a static neuron. On the other hand, $\mathcal{E}_j^{(i)}(\mathcal{N}, \mathcal{D})$ close to $0.89$ (the entropy of a Bernoulli distribution with $p = 0.5$) indicates that the probability of neuron $(i, j)$ being activated or deactivated is around 50% for the data distribution $\mathcal{D}$, suggesting a high degree of uncertainty and strong non-linear representation capacity.

By introducing neuron entropy as a measure, we gain a better understanding of each neuron's contribution to the network's expressive power and their role in enhancing the network's overall non-linear capabilities. As neuron entropy increases, the neural network becomes more capable of capturing complex zones in the data, thereby improving its overall performance.

Figure 1 shows the neuron entropy of the neuron $\hat{a}_j^{(i)}$ where the x-axis is the probability of the zone of $\hat{a}_j^{(i)}$ is 0: $P(\hat{a}_j^{(i)} = 0)$. For activation with two zones (such as ReLU, PReLU), neuron entropy arrives its maximum around 0.89 when $P(\hat{a}_j^{(i)}(\mathbf{x}) = 0) = 1/2$. For activation with 3 zones (such as Sigmoid, ReLU-6), the maximum entropy is 1.59 when $P(\hat{a}_j^{(i)}(\mathbf{x}) = 0) = 1/3$. For activation with 4 zones (such as Swish, Piecewise Linear), the maximum entropy is 1.89 when $P(\hat{a}_j^{(i)}(\mathbf{x}) = 0) = 1/4$. For activation with 5 zones (such as Extended ReLU, Custom Piecewise Function), the maximum entropy is 2.39 when $P(\hat{a}_j^{(i)}(\mathbf{x}) = 0) = 1/5$. And for activation with 6 zones (such as Multi-Slope ReLU, Complex Sigmoid-based Function), the maximum entropy is 2.59 when $P(\hat{a}_j^{(i)}(\mathbf{x}) = 0) = 1/6$.

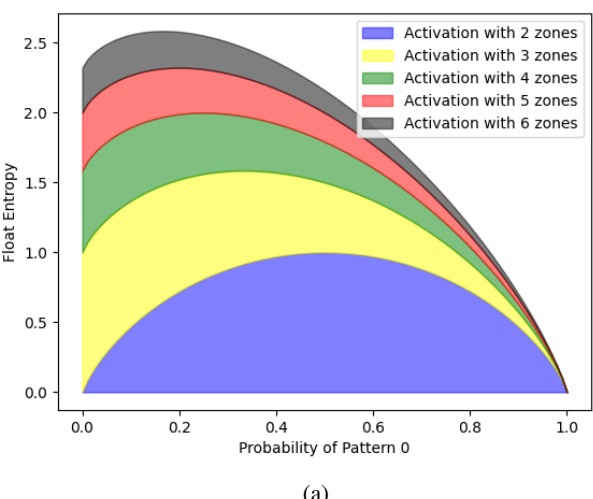

(a)

Figure 1: The dynamic entropy of activation functions with $k$ different patterns given the probability of pattern is 0 (x-axis). Each line shows the maximum value of entropy, while the shadowed zone is the range of entropy from $k = n$ to $k = n - 1$.

## 4.2 UNDERSTANDING NEURAL NETWORKS THROUGH NEURON INSIGHTS

### 4.2.1 MODEL REPRESENTATIONAL CAPACITY

Figure 2 illustrates the neuron entropy in VGG16 networks trained on the CIFAR10 dataset using different batch sizes. Each model underwent 1800 training steps, with neuron entropy evaluated every 30 steps. The ReLU activation function, with a separation of 0, is applied to all networks, resulting in neurons being either activated or not.

Figures 2g and 2h display the training and test accuracy across different models, demonstrating that smaller batch sizes typically yield higher accuracy and more stable performance on both training and testing datasets. To understand the disparity in performance from the perspective of neuron stability, Figures 2a through 2c present the average neuron entropy across different layers, while Figures 2d to 2f show the variance in neuron entropy at each layer.

In the initial layer (layer 1, Figure 2a), neuron entropy is approximately 4.36 across all models, suggesting that most neurons have about a 50% chance of activation for a given dataset. In intermediate layers (layer 6, Figure 2b), neuron entropy initially rises to around 10.76 before gradually declining.

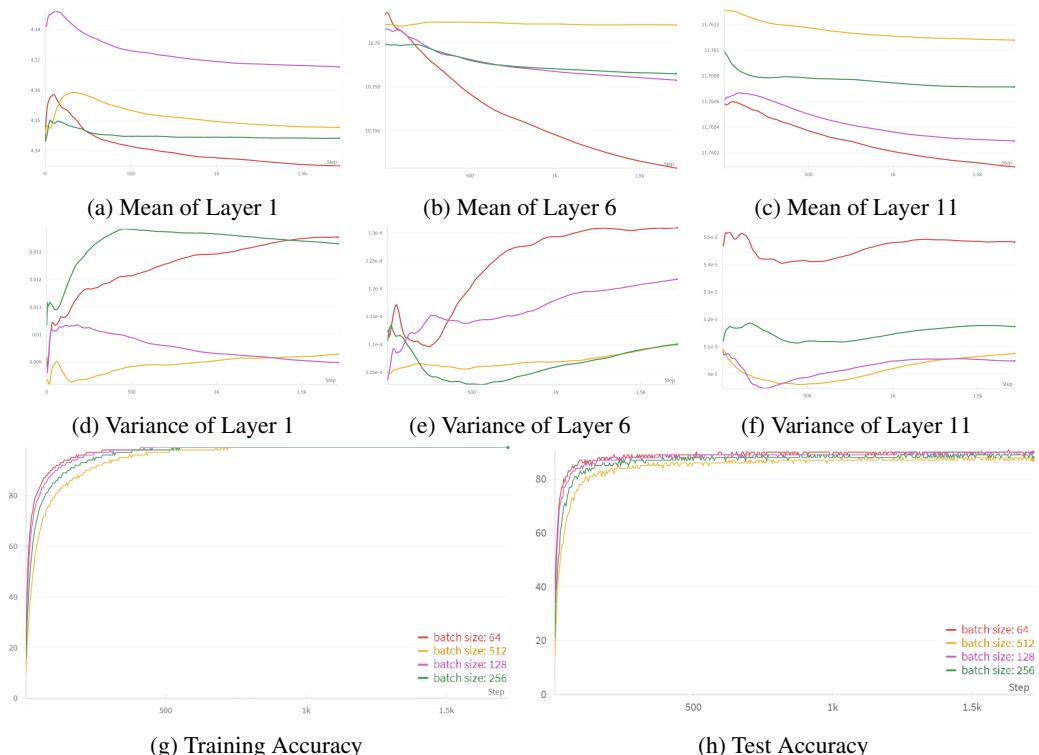

(a) Mean of Layer 1     (b) Mean of Layer 6     (c) Mean of Layer 11

(d) Variance of Layer 1     (e) Variance of Layer 6     (f) Variance of Layer 11

(g) Training Accuracy          (h) Test Accuracy

Figure 2: The mean and variance neuron entropy at different layers for a VGG16 network trained on CIFAR10 dataset.

Notably, neurons in the model with the best performance (batch size 64, red line) exhibit greater stability in these shallow layers.

The most pronounced differences between models are observed in the deeper layers (layer 11, Figure 2c), where higher neuron entropy correlates with improved accuracy on both training and testing datasets. Specifically, in the models with poorer performance, neuron entropy is around 11.761, indicating that most neurons in deep layers remain consistently in the same activation state, regardless of the input. This phenomenon, known as the *dying neuron* issue, leads to a loss of representational capacity Ziping (2023); Lu (2020).

Moreover, when examining the variance of neuron entropy, we observe that the model with the best performance consistently exhibits the lowest variance across different layers. This suggests that neurons in these layers have similar levels of uncertainty.

Analyzing neuron entropy across layers provides valuable insights into neural network performance from multiple perspectives. First, it suggests that a well-performing model tends to have higher neuron entropy in deeper layers and lower entropy in shallower layers. Second, this comparison underscores the distinct roles of each layer. Neurons at the input layer often exhibit instability due to the variability in input images. Previous research indicates that shallow layers are mainly responsible for extracting features from the input, leading to more stable neurons. In contrast, deep layers extract critical information necessary for classification. The increased neuron entropy in deeper layers reflects a more robust capacity to encode features for classification.

### 4.2.2 DYNAMIC NEURON ENTROPY

Figure 3 depicts the evolution of neuron entropy across various layers during training. We track the entropy of neurons throughout the training process at each layer. In the initial layers (e.g., layer 2), most neurons exhibit entropy close to 8.76, indicating high responsiveness to the dataset and capability to distinguish different data instances. During the early stages of training, particularly up to layer 5, neuron entropy increases rapidly and stabilizes at approximately 10.76.

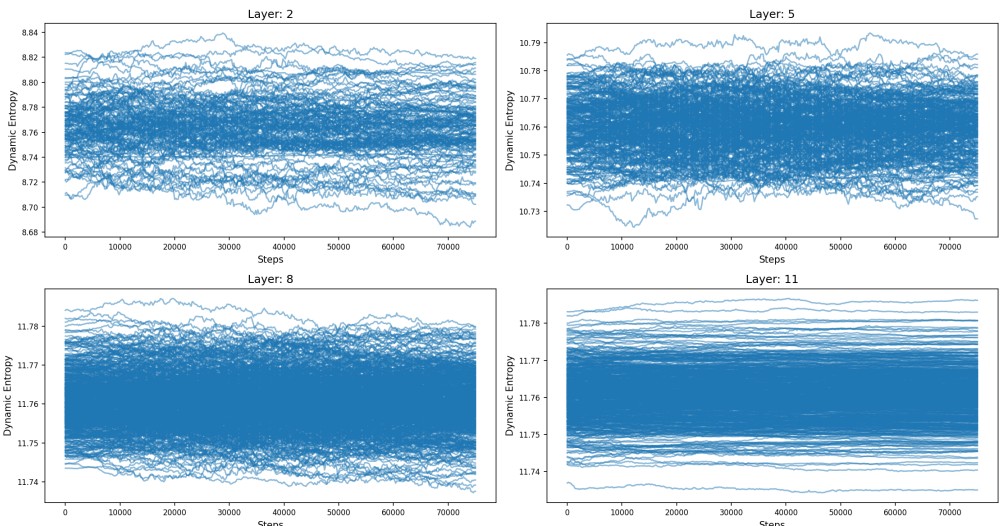

Figure 3: Dynamic of neuron entropy at different layers during training. Each of the lines records the entropy of a neuron during training every 100 steps.

As the model progresses into deeper layers, however, more neurons begin to lose their responsiveness, as observed in layers 8 and 11. For instance, by the end of training, the increment in entropy from layer 5 to layer 8 is roughly 1, which is lower compared to the 2 unit increase observed between layer 1 and layer 5. Additionally, neurons in layer 8 achieve stability in entropy earlier than those in layer 5. At layer 11, the majority of neurons eventually reach a plateau in entropy, indicating a cessation of volatility. This analysis of dynamic neuron entropy provides insight into neuron-specific behavior across the dataset, helping to identify *dying neurons* at various layers—neurons that do not significantly contribute to model performance.

### 4.3 HYBRID ENTROPY-NORM PRUNING (HENP)

#### 4.3.1 WEIGHTED ENTROPY AND HYBRID NORM

In this section, we introduce a novel approach that combines weighted entropy with a hybrid norm for pruning, specifically tailored for convolutional layers and linear layers. This method employs metrics to assess the importance of parameters, making it more suitable for in-depth academic research and theoretical analysis.

Firstly, in the convolutional layer, we propose to calculate the weighted entropy for each filter. Unlike traditional entropy calculations, the entropy for a specific filter $F_i$ is determined by a weighted sum of the entropies of its channels, where the weights are derived from the contribution of each channel to the filter's output:

$$\text{Weighted Entropy}(F_i) = \sum_{c=1}^{C} w_c \times \text{Entropy}(F_i^c)$$

where $C$ represents the number of channels, $F_i^c$ is the $c$-th channel of filter $F_i$, and $w_c$ is the weight assigned to the $c$-th channel based on its significance in the filter's output. This weighted entropy is then combined with a hybrid norm of the filter, which is computed as L2 norm:

$$\text{Pruning Score}(F_i) = f(\text{Weighted Entropy}(F_i), \|F_i\|_{hybrid})$$

where $\|F_i\|_{hybrid}$ represents the hybrid norm, and $f(\cdot)$ is a non-linear function, such as a product or an exponential function, to combine the weighted entropy and hybrid norm. This approach enables a

more refined assessment of each filter's importance by incorporating both local and nuanced channel information.

Secondly, for the linear layer, the method focuses on combining both local and global entropy measures. The local entropy for each output unit is computed first, followed by an aggregation to form a global entropy measure for the entire output layer:

$$\text{Global Entropy}(O) = g\left(\sum_{j=1}^{M} h(p_j) \times \text{Local Entropy}(O_j)\right)$$

where $M$ is the number of output units, $p_j$ is the probability distribution over the outputs, $O_j$ is the $j$-th output unit, and $h(\cdot)$ is a weighting function that adjusts the contribution of each unit's local entropy. The global entropy is then combined with a dynamic hybrid norm of the weight matrix $W$:

$$\text{Pruning Score}(W) = f(\text{Global Entropy}(O), \|W\|_{hybrid})$$

where the function $f(\cdot)$ is similar to that used in the convolutional layer, and the hybrid norm $\|W\|_{hybrid}$ dynamically changes as training progresses. This method allows for a more sophisticated pruning strategy by considering both the local variations and global trends in entropy during the pruning process.

Overall, these methods provide a deeper exploration into the entropy-based pruning strategies by incorporating weighted metrics, hybrid norms, and dynamic adjustments, making them highly applicable for rigorous academic investigations.

## 4.4 HENP ALGORITHM

As discussed above, we propose the neuron entropy metric to evaluate the representational capacity of neural networks and show that a certain amount of *dying neurons* fails to contribute to network predictions. Based on this observation, we introduce our Hybrid Entropy-Norm Pruning method, which eliminates unnecessary parameters from the network by combining norms and neuron entropy. Algorithm 1 in Appendix A.9 outlines the detailed HENP procedure for training and pruning.

At the beginning of training, a binary mask $M$ is initialized, matching the shape of the network parameters. This mask $M$ denotes the neurons that remain active during training. During the forward pass, as shown in step 5 of Algorithm 1, the model computes predictions $\hat{y}$ by applying the masked parameters $\theta \odot M$. Subsequently, step 6 updates the parameters $\theta$ by minimizing the loss function $\mathcal{L}(y, \hat{y})$ using backpropagation. The pruning operation is triggered if the current epoch $t$ matches one of the pruning milestones $P$, as indicated in step 10. In this case, the *Pruning* function is invoked, relying on entropy to select neurons for pruning. An activation zone counter $pc$ is initialized, where $pc^{(i)}j[k]$ represents the occurrence count of zone $k$ for neuron $(i, j)$. For each block $h_i$, the pre-activation $z_i$ is computed at step 17, followed by the activation zone $\hat{a}^{(i)}j$ for each neuron at step 18. The current zone's counter $\hat{a}^{(i)}j$ for neuron $(i, j)$ is incremented. After iterating over the entire dataset, the frequency distribution of each zone estimates the zone distribution for neuron $(i, j)$. The entropy $\mathcal{E}^{(i)}j$ for neuron $(i, j)$ is calculated based on the zone distribution $pc^{(i)}j$. Finally, neurons meeting the pruning criteria are removed by setting the corresponding entries in the mask $M^{(i)}j$ to zero.

For clarity, we separate the training and pruning phases in Algorithm 1. In practice, however, the computation of neuron entropy can be incorporated into the training phase, with the zone counter updates being performed on a batch-wise basis. In Section 5, we explore the impact of different pruning criteria on the performance of pruning methods, taking ResNet18 as a case study.

## 5 EXPERIMENTS

### 5.1 BLEND ENTROPY WITH NORM

As discussed in Section 4.3, for convolutional layers, the neuron entropy and norm are in different shapes. Therefore, an averaged entropy is computed for each filter and fused with the norm. In the following experiments, we consider:

- Norm $\times$ Averaged Layer Entropy (HENP-1),
- Harmonic Mean of Norm and Layer Entropy (HENP-2).

We start by investigating the performance of networks pruned with different methods. Each of the network is initialized with He Normal Distribution He et al. (2015), and trained by SGD optimizer on CIFAR10 dataset with batch size of 128 for 300 epochs with initial learning rate of 0.01. To ensure the stability of pruned parameters and enhance the performance of the remaining weights, the optimizer was configured with a weight decay of $5 \times 10^{-4}$ and a momentum of 0.9. These settings aim to prevent the update of pruned parameters while maintaining the regularization and convergence properties of the optimization process.

**Pruning Details.** In these experiments, the network is pruned at the end of each epoch to allow recovery and optimization of the remaining parameters. The justification for this training procedure can be found in Section 5.3. During each pruning epoch, the entropy of each neuron is recorded, and an importance score is computed by integrating the layer entropy with the model norms at the end of training. Parameters across different layers are then pruned proportionally based on their importance scores.

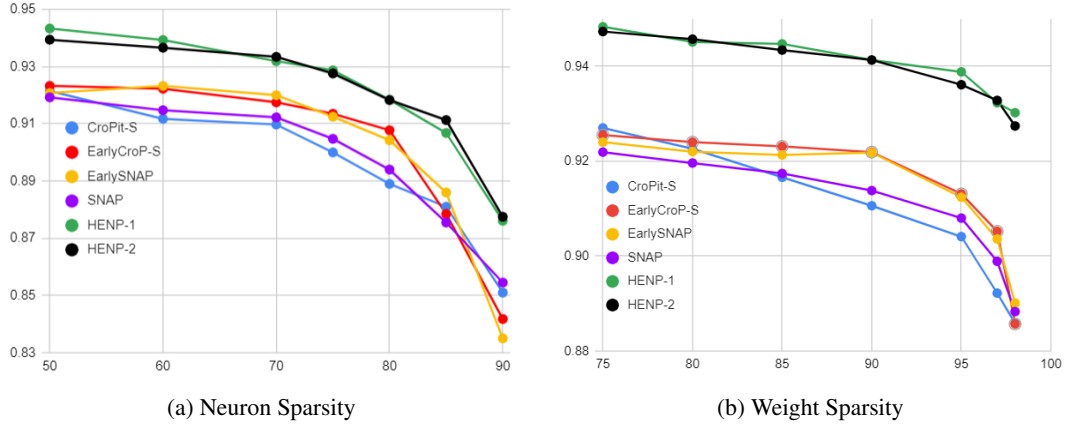

(a) Neuron Sparsity  (b) Weight Sparsity

Figure 4: For ResNet-18 networks on CIFAR-10 trained with ADAM, HENP can find sparser solutions maintaining better performance than other structured approaches. **Left**: Neural sparsity, structured methods. **Right**: Weight sparsity, structured methods.

Our approach is a structured one. Pruning occurs during training, transitioning from dense networks to sparse networks. The methods we compare against also fall within this paradigm, excluding methods such as Lasby et al. Mike (2023), despite their impressive performance, as they are fundamentally non-structured pruning followed by structured recombination. We adopt the following structured pruning baselines: Crop-it/EarlyCrop John (2022), SNAP Stijn (2020), and a modified version of early pruning strategy from Rachwan et al. John (2022) identified as EarlySNAP. These baselines are trained using the recommended configurations from the original authors and are not influenced by the regularization regime employed by our method. In all cases, our approach matches or outperforms other structured pruning methods (Figure 4). And this indicates that incorporating neuron entropy during pruning helps preserve more useful parameters. In essence, neuron entropy serves as a metric for evaluating the network's expressive power.

**Comparison of Pruning Methods.** Figure 4 compares the accuracy of different hybrid pruning networks. Compared to other models, our method, which combines norm and entropy for importance

score pruning, achieves higher accuracy after pruning. Specifically, norm pruning with entropy-normed parameters performs the best, improving the accuracy of the baseline model by 2% to 5% in the sparsity range of 0.75 to 0.98. This suggests that considering neuron entropy during the pruning process helps retain more useful parameters. In other words, neuron entropy can be viewed as an indicator of the network's representation capacity.

## 5.2 LEAKY ReLU

Our method is designed around ReLU activation functions, where neurons can deactivate completely, leading to dying neurons that can be pruned. However, other activation functions like Leaky ReLU Maas (2013) also have a soft saturated zone. We hypothesize that neurons firing solely from this saturated zone contribute minimally to predictions and can be considered nearly dying. To test this, we apply our pruning method to a network with Leaky ReLU activations, removing neurons with only negative activation across a significant minibatch. Once more, our approach surpasses other structured pruning methods, as depicted in Figure 5.

## 5.3 PRUNING FREQUENCY

To understand the effect of pruning towards the models during training, Figure 6 and 7 reports the test accuracy of the pruning method with HENP-1 and HENP-2. The model accuracy drops after each time parameter trim and gradually recovers to during the non-pruning epochs. As model sparsity increases, the model has a higher impairment on performance and finds it harder to recover. This suggests that the pruning frequency can affect the final performance of the network.

Figure 6 and 7 compares the model accuracy given different pruning frequencies under different target sparsity. Given sparsity $S$, models are pruned every 1 2, 5, 10, 15 epochs, where each time the same proportion of parameters is removed from the model. Low pruning frequency implies larger amount each time.

For the models pruned by neurons, increasing the pruning frequency will result in a slightly higher test accuracy. The model with a pruning frequency of 15 demonstrated a higher test accuracy compared to models with other pruning frequencies. But for the models pruned by norms, increasing the pruning frequency results in a slight improvement in test accuracy. Specifically, the model with a pruning frequency of 15 exhibited higher test accuracy compared to models with other pruning frequencies. On the other hand, when introducing the neuron entropy to the importance score, HENP (entropy-norm) models with higher pruning frequency achieves better result. Increasing the pruning frequency from 1 to 15 boosts the model performance by around 2%.

## 6 CONCLUSION

In this study, we introduce a novel analytical framework aimed at elucidating the representation capabilities of neural networks. Building on prior research into the activation zones of neural networks, we present the concepts of dynamic and static neurons to characterize neuron stability. Our approach extends previous analyses from a focus on single activation zones to broader contexts. We specifically employ a neuron entropy metric to track neuron volatility across various scenarios, facilitating a comprehensive examination of neural network expressivity. Finally, we demonstrate the applicability of our findings to downstream tasks.

Moreover, our experiments with Leaky ReLU demonstrate the method's compatibility with activation functions featuring softer saturation zones compared to ReLU. This compatibility suggests potential for sparsifying neural network architectures during training, given their reliance on activation functions such as GELU and Swish. Considering the typical scale of training for these models, our method could yield significant computational and environmental advantages.

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

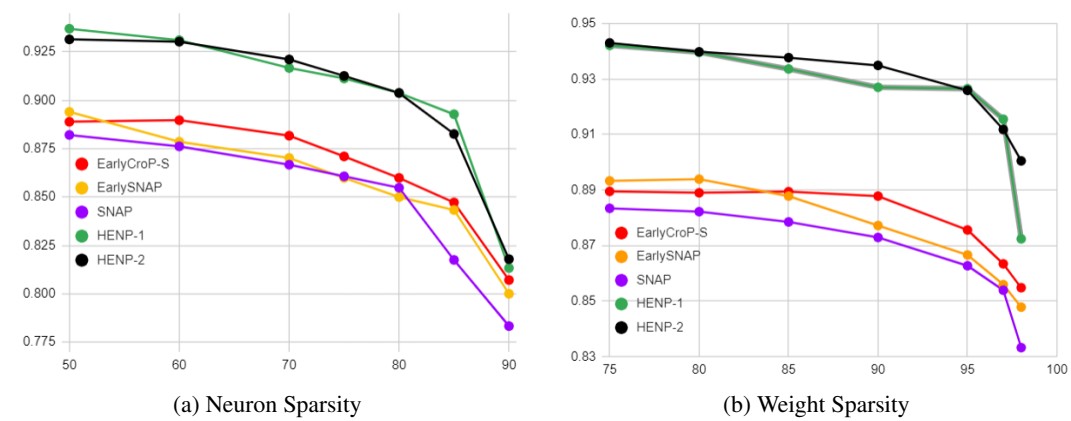

(a) Neuron Sparsity

(b) Weight Sparsity

Figure 5: ResNet-18 networks with *Leaky ReLU* trained on CIFAR-10. HENP again outperforms the baseline structured pruning methods. **Left**: Neural sparsity, structured methods. **Right**: Weight sparsity, structured methods.

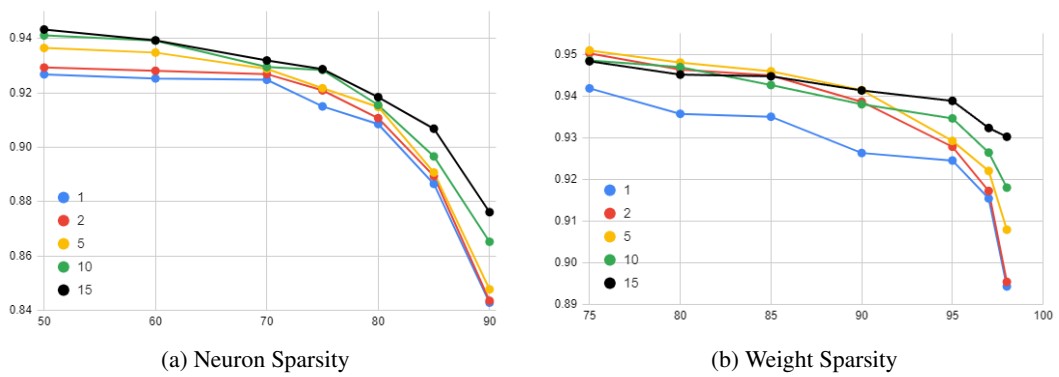

(a) Neuron Sparsity

(b) Weight Sparsity

Figure 6: Comparison of Pruning Frequency with different sparsity of HENP-1.

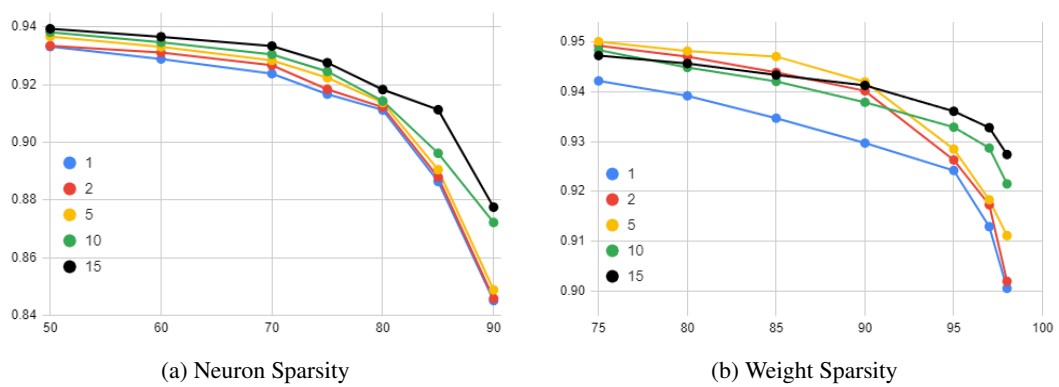

(a) Neuron Sparsity

(b) Weight Sparsity

Figure 7: Comparison of Pruning Frequency with different sparsity of HENP-2.

Sanjeev Arora, et al. Stronger generalization bounds for deep nets via a compression approach. *arXiv preprint arXiv:1802.05296*, 2018.

Y. Bengio, et al. Learning long-term dependencies with gradient descent is difficult. *IEEE Transactions on Neural Networks*, 5(2):157–66, 1994.

Davis Blalock, et al. What is the state of neural network pruning? *arXiv preprint arXiv:2003.03033*, 2020.

Sébastien Bubeck, et al. A law of robustness for two-layers neural networks. *arXiv preprint arXiv:2009.14444*, 2020.

Han Cai, et al. Once-for-all: Train one network and specialize it for efficient deployment. *arXiv preprint arXiv:1908.09791*, 2020.

Rich Caruana, et al. Intelligible models for healthcare: Predicting pneumonia risk and hospital 30-day readmission. *Proceedings of the 21th ACM SIGKDD International Conference on Knowledge Discovery and Data Mining*, pp. 1721–30, 2015.

Chaofan Chen, et al. This looks like that: Deep learning for interpretable image recognition. *arXiv preprint arXiv:1806.10574*, 2019.

Yu Cheng, et al. Model compression and acceleration for deep neural networks: The principles, progress, and challenges. *IEEE Signal Processing Magazine*, 35(1):126–36, 2018.

Setareh Cohan, et al. Understanding the evolution of linear regions in deep reinforcement learning. *NIPS*, 2022.

Finale Doshi-Velez and Been Kim. Towards a rigorous science of interpretable machine learning. *arXiv preprint arXiv:1702.08608*, 2017.

Thomas Elsken, et al. Neural architecture search: A survey. *arXiv preprint arXiv:1808.05377*, 2019.

Amir Gholami, et al. A survey of quantization methods for efficient neural network inference. *arXiv preprint arXiv:2103.13630*, 2021.

Ian Goodfellow, et al. Deep learning. In *MIT Press*, 2017.

Ian J. Goodfellow, et al. Explaining and harnessing adversarial examples. *arXiv preprint arXiv:1412.6572*, 2015.

Jianping Gou, et al. Knowledge distillation: A survey. *International Journal of Computer Vision*, 129(6):1789–819, 2021.

Song Han, et al. Deep compression: Compressing deep neural networks with pruning, trained quantization and huffman coding. *arXiv preprint arXiv:1510.00149*, 2016.

Boris Hanin and Rolnick David. Complexity of linear regions in deep networks. *arXiv preprint arXiv:1901.09021*, 2019.

Kaiming He, Xiangyu Zhang, Shaoqing Ren, and Jian Sun. Delving deep into rectifiers: Surpassing human-level performance on imagenet classification. In *Proceedings of the IEEE international conference on computer vision*, pp. 1026–1034, 2015.

Geoffrey Hinton, et al. Distilling the knowledge in a neural network. *arXiv preprint arXiv:1503.02531*, 2015.

Sepp Hochreiter and Jürgen Schmidhuber. Long short-term memory. *Neural Computation*, 9(8): 1735–80, 1997.

Benoit Jacob, et al. Quantization and training of neural networks for efficient integer-arithmetic-only inference. *2018 IEEE/CVF Conference on Computer Vision and Pattern Recognition*, pp. 2704–13, 2018.

Xiaoheng Jiang, et al. Deep neural networks with elastic rectified linear units for object recognition. *Neurocomputing*, 275:1132–1139, 2018.

Rachwan John, et al. Winning the lottery ahead of time: Efficient early network pruning. *International Conference on Machine Learning, ICML 2022*, pp. 18293–18309, 2022.

Lee Joo Hyung, et al. Jaxpruner: A concise library for sparsity research. *arXiv preprint arXiv:2304.14082*, 2023.

Raghuraman Krishnamoorthi. Quantizing deep convolutional networks for efficient inference: A whitepaper. *arXiv preprint arXiv:1806.08342*, 2018.

Carl Lemaire, et al. Structured pruning of neural networks with budget-aware regularization. *arXiv preprint arXiv:1811.09332*, 2019.

Benjamin Letham, et al. Interpretable classifiers using rules and bayesian analysis: Building a better stroke prediction model. *The Annals of Applied Statistics*, 9(3), 2015.

Zechun Liu, et al. Metapruning: Meta learning for automatic neural network channel pruning. *2019 IEEE/CVF International Conference on Computer Vision (ICCV)*, pp. 3295–304, 2019.

Lu Lu, et al. Dying relu and initialization: Theory and numerical examples. *arXiv preprint arXiv:1903.06733*, 2020.

Andrew L Maas, et al. Rectifier nonlinearities improve neural network acoustic models. *in ICML Workshop on Deep Learning for Audio, Speech and Language Processing*, 2013.

Lasby Mike, et al. Dynamic sparse training with structured sparsity. *CoRR*, 2023.

Grégoire Montavon, et al. Methods for interpreting and understanding deep neural networks. *Digital Signal Processing*, 73:1–15, 2018.

Guido Montúfar, et al. On the number of linear regions of deep neural networks. *arXiv preprint arXiv:1402.1869*, 2014.

W. James Murdoch, et al. Interpretable machine learning: Definitions, methods, and applications. *arXiv preprint arXiv:1901.04592*, 2019.

Razvan Pascanu, et al. On the difficulty of training recurrent neural networks. *arXiv preprint arXiv:1211.5063*, 2013.

Ilija Radosavovic, et al. Designing network design spaces. *2020 IEEE/CVF Conference on Computer Vision and Pattern Recognition (CVPR)*, pp. 10425–33, 2020.

Maithra Raghu, et al. Svcca: Singular vector canonical correlation analysis for deep learning dynamics and interpretability. *arXiv preprint arXiv:1706.05806*, 2017.

Wojciech Samek, et al. Explainable artificial intelligence: Understanding, visualizing and interpreting deep learning models. *arXiv preprint arXiv:1708.08296*, 2017.

Ghada Sokar, et al. The dormant neuron phenomenon in deep reinforcement learning. *International Conference on Machine Learning, ICML 2023, 23-29 July 2023, Honolulu, Hawaii, USA*, 202: 32145–32168, 2023.

Verdenius Stijn, et al. Pruning via iterative ranking of sensitivity statistics. *CoRR*, 2020.

Mukund Sundararajan, et al. Axiomatic attribution for deep networks. *arXiv preprint arXiv:1703.01365*, 2017.

Christian Szegedy, et al. Intriguing properties of neural networks. *arXiv preprint arXiv:1312.6199*, 2013.

Yonglong Tian, et al. Contrastive representation distillation. *arXiv preprint arXiv:1910.10699*, 2022.

Evci Utku, et al. Rigging the lottery: Making all tickets winners. *Proceedings of the 37th International Conference on Machine Learning, ICML 2020*, pp. 2943–2952, 2020.

Tim Whitaker and Darrell Whitley. Synaptic stripping: How pruning can bring dead neurons back to life. *2023 International Joint Conference on Neural Networks (IJCNN)*, 2023.

Glorot Xavier and Yoshua Bengio. Understanding the difficulty of training deep feedforward neural networks. *Proceedings of the Thirteenth International Conference on Artificial Intelligence and Statistics*, 9:249–256, 2010.

Yuhui Xu, et al. Pc-darts: Partial channel connections for memory-efficient architecture search. *arXiv preprint arXiv:1907.05737*, 2020.

Arber Zela, et al. Understanding and robustifying differentiable architecture search. *arXiv preprint arXiv:1909.09656*, 2020.

Chiyuan Zhang, et al. Understanding deep learning (still) requires rethinking generalization. *Communications of the ACM*, 64(3):107–15, 2021.

Yucong Zhou, et al. Learning specialized activation functions with the piecewise linear unit. In *2021 IEEE/CVF International Conference on Computer Vision (ICCV)*, pp. 12075–84, 2021.

Jiang Ziping, et al. Delve into neural activations: Toward understanding dying neurons. *IEEE Transactions on Artificial Intelligence*, 4(4):959–971, 2023.

# A APPENDIX

## A.1 SYMBOLIC REPRESENTATION

Consider a classification task with $c$ different classes. Let $\mathcal{X} \subset \mathbb{R}^d$ represent the input space and $\mathcal{Y} = \{1, 2, \ldots, c\}$ represent the set of class labels. The data distribution is denoted as $\mathcal{D} = \mathcal{D}_\mathbf{x} \times \mathcal{D}_y$ over the input-label pairs $(\mathbf{x}, y) \in \mathcal{X} \times \mathcal{Y}$. Let $\mathcal{N}$ be a feedforward neural network with $d$ blocks, parameterized by $\boldsymbol{\theta}$ (which includes weight matrices and bias vectors). The network defines a mapping $f : \mathbb{R}^d \to \mathbb{R}^c$, which outputs a score vector over the $c$ classes.

Assume that the network is composed of $d$ blocks, such that $f = h_d \circ h_{d-1} \circ \cdots \circ h_1$. For $i \in \{1, 2, \ldots, d-1\}$, each block is defined as $h_i = \pi_i \circ \phi_i$, where $\phi_i(\mathbf{x}) = \|\mathbf{W}^{(i)}\|\mathbf{x} + \|\mathbf{b}^{(i)}\|$. Here, $\|\mathbf{W}^{(i)}\| \in \mathbb{R}^{n_i \times n_{i-1}}$ denotes the norm of the weight matrix, and $\|\mathbf{b}^{(i)}\| \in \mathbb{R}^{n_i}$ denotes the norm of the bias vector. The function $\pi_i$ is the activation function applied element-wise. The final block is defined as $h_d = \phi_d$, where $\phi_d(\mathbf{x}) = \|\mathbf{W}^{(d)}\|\mathbf{x} + \|\mathbf{b}^{(d)}\|$ and $\|\mathbf{W}^{(d)}\| \in \mathbb{R}^{c \times n_{d-1}}$, $\|\mathbf{b}^{(d)}\| \in \mathbb{R}^c$. Note that the last block omits the activation function.

To further clarify, the norm $\|\mathbf{W}^{(i)}\|$ corresponds to a specific norm (e.g., Frobenius norm) of the weight matrix $\mathbf{W}^{(i)}$, and similarly, $\|\mathbf{b}^{(i)}\|$ corresponds to the norm of the bias vector $\mathbf{b}^{(i)}$. The role of these norms is crucial in understanding the scaling behavior of each block, particularly in the context of regularization and generalization in neural networks.

Given an input $\mathbf{x} \in \mathcal{X}$, the network produces a prediction $\hat{y} = \arg\max_{i \in \mathcal{Y}} f_i(\mathbf{x})$, where $f_i(\mathbf{x})$ denotes the score corresponding to class $i$. The overall input-output relationship of the network can be expressed as:

$$f(\mathbf{x}) = \mathbf{W}^{(d)}\pi_{d-1}\left(\mathbf{W}^{(d-1)}\cdots\pi_1\left(\mathbf{W}^{(1)}\mathbf{x} + \mathbf{b}^{(1)}\right)\cdots + \mathbf{b}^{(d-1)}\right) + \mathbf{b}^{(d)}. \qquad (2)$$

For training, the network parameters $\boldsymbol{\theta}$ are optimized to minimize a loss function, typically the cross-entropy loss for classification:

$$\mathcal{L}(\boldsymbol{\theta}; \mathbf{x}, y) = -\log\left(\frac{\exp(f_y(\mathbf{x}))}{\sum_{i=1}^c \exp(f_i(\mathbf{x}))}\right). \qquad (3)$$

To facilitate the analysis of the network, particularly in terms of activation zones, zones, and flows, we introduce the following formalisms:

- $\mathcal{R}^{(i)}$: The set of all possible activation zones at block $i$. Each zone is defined by a unique zone of neuron activations at this block.
- $\mathcal{P}^{(i)}$: The set of all possible activation flows up to block $i$. A flow is defined by the sequence of activation zones from the input block to block $i$.
- $\mathcal{A}^{(i)}(\mathbf{x})$: The activation zone of an input $\mathbf{x}$ at block $i$, represented as a binary vector indicating the neurons that are activated.
- $\pi^{(i)}(\mathbf{x})$: The activation flow for an input $\mathbf{x}$ up to block $i$, which is the sequence of activation zones from block 1 to block $i$.

These formalisms provide a formal framework to describe how different inputs are processed by the network and how the network's internal structure gives rise to distinct activation zones and flows. This foundation is critical for the subsequent analysis of how neural networks partition the input space and how these partitions relate to the network's decision-making process.

## A.2 ZONAL ACTIVATION ZONES AND LOCAL BEHAVIOR ANALYSIS OF NEURAL NETWORKS

Neural networks establish intricate mappings by leveraging layered architectures and nonlinear activation functions. To gain deeper insight into the local dynamics and expressive power of these networks, we present a set of concepts and definitions. Our analysis is systematically organized through precise formalizations and relevant references.

### A.2.1 Neural Activation Signature and Subspace

Given a neural network $\mathcal{N}$ with piecewise linear activation functions, the input domain $\mathbb{R}^n$ is partitioned into multiple zones, within each of which the network's mapping is linear. We define this partitioning formally with the concept of a neural activation signature.

**Definition 4** (Extended Neural Signature and Corresponding Subspace). *Consider an activation function $\sigma(\cdot)$ with breakpoints $\Gamma = \{\gamma_1, \gamma_2, \ldots, \gamma_q\}$ that partition its domain into $q + 1$ intervals $U = \{U_0, U_1, \ldots, U_q\}$. For a neural network $\mathcal{N}$ and an input $\mathbf{x} \in \mathbb{R}^n$, the neural activation signature $\mathcal{A}(\mathbf{x})$ is an indexed set:*

$$\mathcal{A}(\mathbf{x}) = \{a_j^{(i)}(\mathbf{x}) \mid a_j^{(i)}(\mathbf{x}) \in \{0, 1, \ldots, q\}, (i, j) \in \mathcal{I}\},$$

*where $a_j^{(i)}(\mathbf{x})$ indicates the interval $U_k$ that the pre-activation value $z_j^{(i)}(\mathbf{x})$ falls into. The corresponding subspace is defined as:*

$$\mathcal{R}(\mathcal{A}) = \{\mathbf{x} \in \mathbb{R}^n \mid \mathcal{A}(\mathbf{x}) = \mathcal{A}\}.$$

For instance, the ReLU activation function $\sigma(x) = \max(0, x)$ divides the input space into an active zone $\mathbb{R}_+$ and an inactive zone $\mathbb{R}_-$. The tanh function can be similarly partitioned using breakpoints $\{-1, 1\}$.

### A.2.2 Transition Across Adjacent Subspaces

As the network complexity increases, the zones defined by a complete neural activation signature become smaller. To study behavior over larger input spaces, we introduce the concept of partial neural signatures.

**Definition 5** (Partial Neural Signature and Aggregated Subspace). *A partial neural signature $\mathcal{A}_{partial}(\mathbf{x})$ is defined by leaving some neurons' activation states unspecified. The corresponding subspace is then a union of multiple complete activation zones:*

$$\mathcal{R}(\mathcal{A}_{partial}) = \bigcup_j \mathcal{R}(\mathcal{A}_j),$$

*where each $\mathcal{A}_j$ is a complete neural activation signature consistent with $\mathcal{A}_{partial}$.*

This concept allows us to extend our analysis to broader input zones, capturing the network's expressive power over these larger spaces.

### A.2.3 Neural Activation Flow

To further analyze the local behavior of neural networks, we introduce the notion of a neural activation flow, which tracks the sequence of activated neurons from input to output.

**Definition 6** (Neural Activation flow). *A neural activation flow $\tau(\mathbf{x})$ is a sequence of neurons that defines a flow through the network from the input to the output. Formally, it is represented as:*

$$\tau(\mathbf{x}) = \{(i_1, j_1), (i_2, j_2), \ldots, (i_m, j_m)\},$$

*where $(i_k, j_k)$ denotes the activated neuron in layer $k$ along this flow.*

This subsection formalizes the concepts of neural activation signatures, neural trajectories, and the consistency of neuron states and trajectories. These tools provide a structured framework for analyzing the local behavior of neural networks and lay the groundwork for deeper exploration in subsequent sections on geometric analysis and flow decomposition.

### A.3 Spatial Understanding

In this section, we explore the connection between activation zones and zones in a neural network from a spatial perspective. We present new lemmas that illustrate how the input domain is partitioned into activation zones and discuss the formal properties of these zones.

### A.3.1   CONVEX ACTIVATION ZONE

To understand the concept of activation zones, consider a neural network $\mathcal{N}$ with a monotonic activation function $\pi$. Given an input space $\mathbb{R}^n$, the generalized activation zone describes the activation status of each neuron at the intermediate layers. The following lemma formalizes the convexity of activation zones corresponding to each activation zone.

**Lemma 2** (Convex Activation Zone Lemma). *Let $\mathcal{N}$ be a neural network with a monotonic activation function $\pi$. Then, for any activation zone $\mathcal{A}$, the corresponding activation zone $\mathcal{C}(\mathcal{A})$ is a convex set. In other words, each activation zone $\mathcal{A}$ uniquely determines a convex activation zone $\mathcal{C}(\mathcal{A})$.*

This lemma implies that the mapping from activation zones $\{\mathcal{C}\}$ to activation zones $\{\mathcal{A}\}$ is injective. The convexity of these zones provides insights into how the network's neurons react to inputs within different zones of the input space.

### A.3.2   INPUT SPACE COVERAGE

Next, we consider the coverage of the input space by these convex activation zones. The following lemma describes how the input space can be fully partitioned by these zones and certain curved separating surfaces.

**Lemma 3** (Input Space Coverage Lemma). *Let $\mathcal{N}$ be a neural network. The input space $\mathbb{R}^n$ can be represented as the union of all convex activation zones $\mathcal{C}(\mathcal{A})$, or as the union of certain curved separating surfaces $\mathcal{H}$. Specifically, the input space $\mathbb{R}^n$ is fully covered by these activation zones and separating surfaces:*

$$\mathbb{R}^n = \left[ \bigcup_{\mathcal{A}} \mathcal{C}(\mathcal{A}) \right] \bigcup \left[ \bigcup_{\forall i,j,k} \mathcal{H}_{ijk} \right] \tag{4}$$

This lemma shows that every point in the input space either lies within a convex activation zone or on a separating surface, indicating the comprehensive coverage provided by these zones.

### A.3.3   ALMOST SURE INPUT DISTRIBUTION

To further analyze the distribution of inputs within these activation zones, we consider the probability of an input point lying on a separating surface. The following lemma addresses this situation under the assumption that the input distribution has no atoms.

**Lemma 4** (Almost Sure Input Distribution Lemma). *Assume the input distribution $\mathbb{D}$ has no atoms (i.e., it assigns zero probability to single points). Then the probability that an input point $\mathbf{x}$ lies exactly on any curved separating surface $\mathcal{H}$ is zero. Thus, almost every input $\mathbf{x}$ lies within some convex activation zone $\mathcal{C}(\mathcal{A})$.*

This result implies that, under typical conditions, the input space is almost surely covered by the convex activation zones, and the contribution of points lying on separating surfaces is negligible.

### A.3.4   ADJACENT ZONES AND CRITICAL NEURONS

We now consider the relationship between adjacent convex activation zones. The following lemma identifies the neurons responsible for the difference in activation zones between two adjacent zones.

**Lemma 5** (Adjacent Zones Difference Lemma). *For any two input points $\mathbf{x}$ and $\mathbf{x}'$ that are located in adjacent convex activation zones $\mathcal{C}$ and $\mathcal{C}'$, there exists a unique critical neuron $(i, j)$ such that the activation zone of this neuron differs between $\mathbf{x}$ and $\mathbf{x}'$:*

$$\exists!(i,j), \hat{a}_j^{(i)}(\mathbf{x}) \neq \hat{a}_j^{(i)}(\mathbf{x}') \tag{5}$$

This lemma reveals that the difference in activation zones between adjacent zones is caused by a unique critical neuron, while all other neurons maintain the same activation status.

A.4 FLOW-BASED ANALYSIS OF NEURAL NETWORK COMPUTATIONAL STRUCTURE

**Rationale.** The exploration of neural network computational structures can be enhanced by analyzing the flow-dependent decomposition of its piecewise linear mappings. Given a neural network $\mathcal{N}$ with a piecewise linear activation function, the output mapping function $f : \mathcal{R}^{n_0} \to \mathcal{R}^c$ can be decomposed into a combination of flows, each representing an independent computational route from input to output. This section details the formal derivation of such a decomposition and its implications.

**Flow Analysis of Piecewise Linear Mappings** Let $f(\mathbf{x})$ denote the output function of the neural network $\mathcal{N}$, which is assumed to be piecewise linear. The network's output can be represented as a sum of functions corresponding to distinct flows $\mathcal{P}$:

$$f(\mathbf{x}) = \sum_{p \in \mathcal{P}} f_p(\mathbf{x}), \tag{6}$$

where $\mathcal{P}$ is the set of all possible flows through the network, and each function $f_p(\mathbf{x})$ corresponds to the contribution of flow $p$. The decomposition is determined by the activation zone $\sigma(\mathbf{x})$ and the norm $\|\mathbf{w}_p\|$ associated with the flow, where $\|\mathbf{w}_p\|$ represents the norm of the weights along the flow $p$. Together, these components define the contribution of each flow $p$ to the network's output.

**Lemma 6** (Flow-Dependent Output Decomposition). *Given a neural network $\mathcal{N}$ and an input $\mathbf{x}$, the output can be expressed as:*

$$f(\mathbf{x}) = \sum_{p \in \mathcal{P}} \|\mathbf{w}_p\| \cdot \sigma_p(\mathbf{x}) \cdot \mathbf{x}, \tag{7}$$

*where $\sigma_p(\mathbf{x})$ denotes the activation status of flow $p$ under input $\mathbf{x}$, and $\|\mathbf{w}_p\|$ represents the norm of the weight vector along flow $p$. The flow contribution depends on both the input $\mathbf{x}$ and the specific activation zone $\sigma(\mathbf{x})$ triggered within the network.*

**Difference Analysis of Local Computational Zones** The output variation between neighboring input zones can be analyzed through the differences in their corresponding flows. Let $\mathbf{x}$ and $\mathbf{x}'$ be two neighboring inputs, and let the output difference $\Delta f$ be defined as:

$$\Delta f = f(\mathbf{x}') - f(\mathbf{x}). \tag{8}$$

**Theorem 1** (Local Zone Output Difference). *Given the network $\mathcal{N}$, the output difference $\Delta f$ between two neighboring inputs $\mathbf{x}$ and $\mathbf{x}'$ can be expressed as:*

$$\Delta f = \mathbf{J}_f(\mathbf{x}) \cdot (\mathbf{x}' - \mathbf{x}) + \sum_{p \in \mathcal{P}} \Delta \|\mathbf{v}_p\| \cdot \sigma_p(\mathbf{x}), \tag{9}$$

*where $\mathbf{J}_f(\mathbf{x})$ is the Jacobian matrix of the network at input $\mathbf{x}$, and $\Delta\|\mathbf{v}_p\|$ represents the change in the norm of flow vectors between the two inputs. The term $\mathbf{J}_f(\mathbf{x}) \cdot (\mathbf{x}' - \mathbf{x})$ captures the linear approximation of output difference, while the summation term accounts for the nonlinear flow-dependent differences.*

The analysis shows that for two close input points $\mathbf{x}$ and $\mathbf{x}'$, the difference in their outputs can be largely attributed to the flows that change their activation status between these points. This provides a norm-based way of understanding the network's computational behavior in a localized input zone by examining the stability and variability of specific flows.

A.5 EXPRESSIVE POWER OF NEURAL NETWORKS AND NEURON ENTROPY RATIONALE

The expressive power of neural networks largely depends on the activation behavior of the neurons. Theorem 1 suggests that the difference between the outputs of a neural network for different inputs can be decomposed into a static part, which is a (semi-)linear transformation, and a dynamic part, which introduces non-linearity. If the neurons in a neural network remain static in their activation over the support set of the input data distribution $\mathcal{D}$, meaning their output does not change with varying inputs, the network's function would degrade to a linear transformation, significantly reducing its capacity to handle complex tasks. This indicates that a network with an insufficient number

of dynamic neurons has a limited ability to approximate functions with high complexity. As the number of dynamic neurons increases, the representational capacity of the network also increases.

Motivated by the above intuition, we introduce the concept of neuron entropy to better capture the diversity of neuron activation. Neuron entropy quantifies the uncertainty in a neuron's activation zones. Specifically, higher entropy indicates greater uncertainty in the neuron's activations, leading to stronger non-linear representation capacity, and thus enhancing the overall expressive power of the network.

### A.6 NEURON ENTROPY

We establish a rigorous connection between connection diversity and model representational capacity. Let $(q, c)$ and $(q', c')$ be two samples drawn from a dataset $S$, where $c \neq c'$. For the model $P$ to correctly classify these samples, it must distinguish between the inputs $q$ and $q'$ with high accuracy. We introduce connection diversity as a crucial factor that impacts the model's capability in differentiating these inputs.

The difference in model outputs for the two samples, $h(q) - h(q')$, can be dissected into linear and nonlinear components. The linear component is dictated by the local connection structure at $q$, which can be analyzed through a connection matrix that captures the model's orientation. The nonlinear component, on the other hand, is influenced by the number of activated connection flows (ACFs) between the sample pairs $\{q, q'\}$. As the number of ACFs increases, so does the complexity of the function difference, $h(q) - h(q')$, implying enhanced model representation capacity.

As the number of activated connection flows (ACFs) increases, the complexity of the function difference $h(q) - h(q')$ also increases. This enhancement in complexity suggests that the model is capable of capturing more intricate nonlinear relationships between inputs, which is critical for improving classification accuracy.

We extend the analysis from individual sample pairs to the entire dataset $S$. Connection diversity is formalized as a metric that assesses the expected nonlinearity introduced by the model across the dataset. It reflects the model's capacity to navigate complex decision boundaries by evaluating the overall distribution of activated connection flows. The higher the connection diversity, the greater the model's ability to differentiate between diverse sample pairs.

To establish a formal connection between model representational capacity and connection diversity, we introduce the following theorem:

**Theorem 2.** *Consider two models, $P$ and $P'$, which share the same architecture but have different parameter sets. The following assumptions hold:*

1. *The dataset $S$ is continuous, implying that the probability of any specific input vector occurring is zero.*

2. *The parameter sets of both models are sampled from the same distribution.*

3. *Model $P$ has lower connection diversity than model $P'$.*

*Then, the expected difference in representational capacity between the models can be formalized as:*
$$\mathbb{E}(h(q; \theta) - h(q'; \theta)) \leq \mathbb{E}(h(q; \theta') - h(q'; \theta'))$$
*where $\theta$ and $\theta'$ are the respective parameter sets for models $P$ and $P'$.*

This theorem suggests that, under the given assumptions, model $P$, which exhibits lower connection diversity, is less effective at distinguishing between two sample points than model $P'$. Thus, a higher connection diversity implies a greater representational capacity.

We extend our discussion to multiclass classification tasks. In scenarios involving multiple classes, models that can better differentiate between various classes generally exhibit higher complexity. This observation aligns with the importance of connection diversity, as models with greater connection diversity are better equipped to handle the increased complexity inherent in multiclass problems.

Subsequent sections will present empirical results to validate the proposed metric of connection diversity. These experiments will demonstrate its effectiveness in predicting and analyzing model performance across various tasks, underscoring its relevance in practical applications.

### A.7 RELATIONSHIP BETWEEN MODEL EXPRESSIVENESS AND NEURON ACTIVATION

This section rigorously examines the relationship between the expressiveness of a model and the activation of neurons. We formalize this relationship through several theorems that connect neuron activation with key metrics of model expressiveness, such as the number of linear zones, model stability, and zone similarity.

First, let $\mathcal{N}$ be a neural network with piecewise linear characteristics. The input space $\mathcal{X}$ is partitioned into multiple linear zones, where the network's mapping remains linear within each zone. Let #Linear Zones($\mathcal{N}$) denote the number of these zones. The complexity of the model can thus be quantified by the number of linear zones. We propose the following theorem:

**Theorem 3.** *Let $\mathcal{N}, \mathcal{N}'$ be neural networks with identical architectures, and let $\theta, \theta'$ represent their respective parameter sets. If the average neuron activation $\sum_{i,j} \mathcal{A}_j^{(i)}(\theta, \mathcal{D}) > \sum_{i,j} \mathcal{A}_j^{(i)}(\theta', \mathcal{D})$, then #Linear Zones($\mathcal{N}$) > #Linear Zones($\mathcal{N}'$).*

The proof of Theorem 3 builds upon the intuition that higher average neuron activation implies more variability in activation zones across different inputs, which in turn leads to a finer partitioning of the input space into linear zones.

Next, we consider model stability along a flow $\gamma(t)$ in the input space, where $t \in [0, 1]$. The stability can be evaluated by the number of linear zones crossed by $\gamma(t)$. We define the transition density $\tau(\mathcal{N}, \gamma)$ as the number of such zones. The following theorem formalizes the relationship between neuron activation and model stability:

**Theorem 4.** *Let $\mathcal{N}, \mathcal{N}'$ be neural networks as defined above. If $\sum_{i,j} \mathcal{A}_j^{(i)}(\theta, \mathcal{D}) > \sum_{i,j} \mathcal{A}_j^{(i)}(\theta', \mathcal{D})$, then $\tau(\mathcal{N}, \gamma) > \tau(\mathcal{N}', \gamma)$ for a given flow $\gamma(t)$.*

Theorem 4 suggests that networks with higher neuron activation exhibit less stability along certain flows, as they cross more linear zones.

Lastly, we discuss zone similarity, denoted as $PS(\mathcal{D}; \theta)$, which measures the ratio of neurons that exhibit similar activation zones for different inputs from the distribution $\mathcal{D}$. Higher zone similarity generally indicates a reduction in model expressiveness, often leading to phenomena such as dying neurons, where units respond uniformly regardless of input. We state the following theorem:

**Theorem 5.** *Let $\mathcal{N}, \mathcal{N}'$ be neural networks with the same architecture. Given the data distribution $\mathcal{D}$, if $\sum_{i,j} \mathcal{A}_j^{(i)}(\theta, \mathcal{D}) > \sum_{i,j} \mathcal{A}_j^{(i)}(\theta', \mathcal{D})$, then $PS(\mathcal{D}; \theta) > PS(\mathcal{D}; \theta')$.*

Theorem 5 demonstrates that increased neuron activation is associated with greater zone similarity. This heightened similarity could suggest a reduction in expressiveness.

These theorems establish a rigorous basis for examining how neuron activation is connected to essential metrics of model expressiveness. The influence of neuron activation extends beyond merely affecting the number of linear regions and transition density; it also plays a crucial role in determining zone similarity. This provides a detailed framework for evaluating model performance at the level of individual units.

### A.8 PRUNING METHOD RATIONALE

The experiments presented in Section 4.2.1 examine the representational capabilities of neural networks from multiple perspectives. The findings indicate that, particularly in deeper layers of a well-trained model, a significant number of neurons exhibit minimal entropy. This suggests that these neurons consistently transmit the same signals, regardless of the input, thereby contributing little to the classification task.

This observation aligns with earlier discussions on the representational limitations of neural networks Hanin & David (2019); Ziping (2023), highlighting that although neural networks have the potential to model highly complex functions, their actual representational capacity is constrained by inactive neurons, often referred to as *dying neurons*.

Table 1: For ResNet-18 networks on CIFAR-10 trained with ADAM, HENP can find sparser solutions maintaining better performance than other structured approaches. **Neural sparsity**, structured methods.

|  | 50 | 60 | 70 | 75 | 80 | 85 | 90 |
|---|---|---|---|---|---|---|---|
| CroPit-S | 0.9215 | 0.91175 | 0.90975 | 0.9 | 0.889 | 0.881 | 0.851 |
| EarlyCroP-S | 0.92325 | 0.92225 | 0.9175 | 0.9135 | 0.90775 | 0.8785 | 0.84175 |
| EarlySNAP | 0.92075 | 0.92325 | 0.92 | 0.9125 | 0.90425 | 0.886 | 0.835 |
| SNAP | 0.91925 | 0.91475 | 0.91225 | 0.90475 | 0.894 | 0.8755 | 0.8545 |
| **HENP-1** | **0.9433** | **0.9393** | **0.9319** | **0.9287** | **0.9184** | **0.9068** | **0.8761** |
| **HENP-2** | **0.9394** | **0.9366** | **0.9334** | **0.9276** | **0.9183** | **0.9113** | **0.8775** |

To address this issue, we propose HENP (Hybrid Entropy-Norm Pruning) as a method to reduce network size while preserving performance. HENP aims to prune the *dying neurons*, which have minimal impact on the network's overall representational capacity.

## A.9 HENP TRAINING ALGORITHM

---
**Algorithm 1** HENP training algorithm
---
**Input:** Training data $D_{\text{train}}$, Validation data $D_{\text{val}}$, Learning rate $\eta$, Sparsity $S$, Number of epochs $E$

1: **Function** TRAIN$(f, \theta)$
2:     Initialize mask $M \leftarrow 1$
3:     **for** epoch $= 1$ to $E$ **do**
4:         **for** $(x, y) \in D_{\text{train}}$ **do**
5:             $\hat{y} \leftarrow f(x; M \cdot \theta)$                      ▷ Forward pass
6:             Compute loss: $L = \mathcal{L}(\hat{y}, y)$
7:             Backpropagate: $\theta \leftarrow \theta - \eta \frac{\partial L}{\partial \theta}$
8:         **end for**
9:         Compute training accuracy $A_{\text{train}}$
10:        **if** pruning condition is met **then**
11:            $M \leftarrow$ PRUNING$(\theta, L)$                  ▷ Pruning step
12:        **end if**
13:    **end for**
14:    **return** $\theta$
15: **end**
16: **Function** PRUNING$(\theta, L)$
17:    Initialize importance scores $I(\theta) \leftarrow 0$
18:    **for** layer $l$ in model **do**
19:        **for** channel $c$ in layer $l$ **do**
20:            Compute entropy $\mathcal{H}(\theta_c) = -\sum p(\theta_c) \log p(\theta_c)$
21:            Compute norm $\|\theta_c\| = \sqrt{\sum_i \theta_{c,i}^2}$
22:            $I(\theta_c) \leftarrow \eta \cdot \mathcal{H}(\theta_c) \cdot \|\theta_c\|$              ▷ Importance score
23:        **end for**
24:        Calculate threshold $\tau$ for sparsity $S$
25:        Pruning weights: $\theta_l \leftarrow \theta_l \cdot \mathbb{I}(I(\theta_l) > \tau)$
26:    **end for**
27:    **return** $M$
28: **end**
29: **Output:** Pruned model parameters $\theta$
---

Table 2: For ResNet-18 networks on CIFAR-10 trained with ADAM, HENP can find sparser solutions maintaining better performance than other structured approaches. **Weight sparsity**, structured methods.

|  | 75 | 80 | 85 | 90 | 95 | 97 | 98 |
|---|---|---|---|---|---|---|---|
| CroPit-S | 0.927 | 0.9226 | 0.9166 | 0.9106 | 0.9041 | 0.8922 | 0.8858 |
| EarlyCroP-S | 0.9255 | 0.924 | 0.9231 | 0.9219 | 0.9131 | 0.9052 | 0.8857 |
| EarlySNAP | 0.924 | 0.922 | 0.9213 | 0.9218 | 0.9124 | 0.9036 | 0.8901 |
| SNAP | 0.9219 | 0.9196 | 0.9174 | 0.9138 | 0.908 | 0.8989 | 0.8883 |
| **HENP-1** | **0.9483** | **0.9451** | **0.9447** | **0.9413** | **0.9388** | **0.9323** | **0.9302** |
| **HENP-2** | **0.9473** | **0.9457** | **0.9434** | **0.9413** | **0.9361** | **0.9328** | **0.9274** |

Table 3: ResNet-18 networks with *Leaky ReLU* trained on CIFAR-10. HENP again outperforms the baseline structured pruning methods. **Neural sparsity**, structured methods.

|  | 50 | 60 | 70 | 75 | 80 | 85 | 90 |
|---|---|---|---|---|---|---|---|
| EarlyCroP-S | 0.8889 | 0.8897 | 0.8817 | 0.871 | 0.8599 | 0.8472 | 0.8071 |
| EarlySNAP | 0.894 | 0.8786 | 0.8702 | 0.8599 | 0.85 | 0.8433 | 0.8 |
| SNAP | 0.8821 | 0.8762 | 0.8667 | 0.8607 | 0.8548 | 0.8175 | 0.7833 |
| **HENP-1** | **0.937** | **0.9311** | **0.9167** | **0.9113** | **0.9038** | **0.8928** | **0.8133** |
| **HENP-2** | **0.9315** | **0.9303** | **0.9211** | **0.9126** | **0.9038** | **0.8826** | **0.8179** |

Table 4: ResNet-18 networks with *Leaky ReLU* trained on CIFAR-10. HENP again outperforms the baseline structured pruning methods. **Weight sparsity**, structured methods.

|  | 75 | 80 | 85 | 90 | 95 | 97 | 98 |
|---|---|---|---|---|---|---|---|
| EarlyCroP-S | 0.8895 | 0.889 | 0.8894 | 0.8878 | 0.8756 | 0.8634 | 0.8548 |
| EarlySNAP | 0.8933 | 0.8939 | 0.8878 | 0.8772 | 0.8666 | 0.8559 | 0.8478 |
| SNAP | 0.8834 | 0.8822 | 0.8785 | 0.8729 | 0.8627 | 0.8539 | 0.8332 |
| **HENP-1** | **0.9421** | **0.9396** | **0.9336** | **0.927** | **0.9265** | **0.9155** | **0.8724** |
| **HENP-2** | **0.943** | **0.9398** | **0.9377** | **0.9349** | **0.9259** | **0.9118** | **0.9005** |

Table 5: Comparison of Pruning Frequency (PF) with Neuron Sparsity of HENP-1. The bold values represent the best performance, and the underlined values represent the second-best performance.

| PF | 50 | 60 | 70 | 75 | 80 | 85 | 90 |
|---|---|---|---|---|---|---|---|
| 1 | 0.9268 | 0.9253 | 0.9248 | 0.915 | 0.9085 | 0.8865 | 0.8428 |
| 2 | 0.9293 | 0.9281 | 0.9269 | 0.9209 | 0.9107 | 0.8894 | 0.8435 |
| 5 | 0.9365 | 0.9348 | 0.9288 | 0.9217 | 0.9148 | 0.8907 | 0.8477 |
| 10 | _0.9411_ | _0.9392_ | _0.9295_ | _0.9284_ | _0.9155_ | _0.8966_ | _0.8652_ |
| 15 | **0.9433** | **0.9393** | **0.9319** | **0.9287** | **0.9184** | **0.9068** | **0.8761** |

Table 6: Comparison of Pruning Frequency (PF) with Weight Sparsity of HENP-1.

| PF | 75 | 80 | 85 | 90 | 95 | 97 | 98 |
|---|---|---|---|---|---|---|---|
| 1 | 0.9418 | 0.9357 | 0.935 | 0.9263 | 0.9245 | 0.9154 | 0.8943 |
| 2 | _0.9502_ | 0.9463 | _0.9449_ | 0.9386 | 0.9278 | 0.9172 | 0.8954 |
| 5 | **0.9509** | **0.948** | **0.9459** | **0.9414** | 0.9292 | 0.922 | 0.9079 |
| 10 | 0.9485 | _0.9469_ | 0.9426 | 0.938 | _0.9346_ | _0.9264_ | _0.918_ |
| 15 | 0.9483 | 0.9451 | 0.9447 | _0.9413_ | **0.9388** | **0.9323** | **0.9302** |

Table 7: Comparison of Pruning Frequency (PF) with Neuron Sparsity of HENP-2.

| PF | 50 | 60 | 70 | 75 | 80 | 85 | 90 |
|----|-----|-----|-----|-----|-----|-----|-----|
| 1 | 0.9332 | 0.9289 | 0.9238 | 0.9167 | 0.9112 | 0.8864 | 0.8451 |
| 2 | 0.9335 | 0.9311 | 0.9267 | 0.9184 | 0.9122 | 0.8879 | 0.8458 |
| 5 | 0.9367 | 0.9331 | 0.9284 | 0.9224 | 0.9138 | 0.8905 | 0.8488 |
| 10 | 0.9382 | 0.9347 | 0.9305 | 0.9246 | 0.9143 | 0.8962 | 0.8722 |
| 15 | **0.9394** | **0.9366** | **0.9334** | **0.9276** | **0.9183** | **0.9113** | **0.8775** |

Table 8: Comparison of Pruning Frequency (PF) with Weight Sparsity of HENP-2.

| PF | 75 | 80 | 85 | 90 | 95 | 97 | 98 |
|----|-----|-----|-----|-----|-----|-----|-----|
| 1 | 0.9422 | 0.9392 | 0.9347 | 0.9297 | 0.9242 | 0.9129 | 0.9005 |
| 2 | 0.9493 | 0.9471 | 0.9439 | 0.9402 | 0.9263 | 0.9173 | 0.9019 |
| 5 | **0.9501** | **0.9482** | **0.9471** | **0.942** | 0.9285 | 0.9183 | 0.9111 |
| 10 | 0.9484 | 0.9449 | 0.9421 | 0.9379 | 0.9329 | 0.9287 | 0.9215 |
| 15 | 0.9473 | 0.9457 | 0.9434 | 0.9413 | **0.9361** | **0.9328** | **0.9274** |

