# OpenReview forum: "HENP: Dynamic Pruning via Neuron Entropy"
_ICLR.cc/2025/Conference — ICLR 2025 Conference Withdrawn Submission_

### Official Review · Reviewer_2ZXG · 2024-11-03

**Soundness:** 2
**Presentation:** 1
**Contribution:** 1
**Rating:** 3
**Confidence:** 4

**Summary:**

In this work, the authors propose a method for pruning neurons through the employment of entropy estimations for multi-slope ReLU activations. The paper contributes with a qualitative definition of static and dynamic neurons, and with blending the entropy estimation with the average norm of their activation. The empirical evaluation is performed on VGG16/ResNet-18 on CIFAR-10.

**Strengths:**

- Structured pruning approaches are important to improve computational complexity at the inference time of the models.
- Overall the idea is sound.

**Weaknesses:**

- The idea of employing entropy to guide pruning is not new, as other works already employ a neuron entropy approach to guide pruning [A-C]. The technical novelty seems limited.
- The empirical evaluation is weak, in terms of datasets/architectures variety and comparisons.
- The writing is in general unclear or even straightforward in many sections (eg. in Sec. 4.1, eq. (1) has many undefined symbols, and the peak in the entropy for many rectified zones is obvious). Besides, some sections, like 4.2.1, are too specific detailing some quantitative experiments (which could be moved to the experiment section).

[A] Spadaro, Gabriele, et al. "Shannon strikes again! entropy-based pruning in deep neural networks for transfer learning under extreme memory and computation budgets." Proceedings of the IEEE/CVF International Conference on Computer Vision. 2023.

[B] Liao, Zhu, et al. "Can Unstructured Pruning Reduce the Depth in Deep Neural Networks?." Proceedings of the IEEE/CVF International Conference on Computer Vision. 2023.

[C] Quétu, Victor, et al. "The simpler the better: An entropy-based importance metric to reduce neural networks’ depth." Joint European Conference on Machine Learning and Knowledge Discovery in Databases. Cham: Springer Nature Switzerland, 2024.

**Questions:**

- How is the approach performing on larger datasets like ImageNet-1k? What about more recent architectures like transformers (eg. SWIN)?
- How is the method comparing with other entropy-based pruning methods like [A-C] ?
- How is the proposed approach compared to more recent pruning approaches, like [D-G] ?
- What is the role played by the hyper-parameters in the proposed method ?

[D] G. Fang, X. Ma, M. Song, M. B. Mi, and X. Wang, “Depgraph: Towards any structural pruning,” in Proceedings of the IEEE/CVF Conference on Computer Vision and Pattern Recognition, 2023, pp. 16 091–16 101.

[E] Park, Jun-Hyung, et al. "Dynamic structure pruning for compressing CNNs." Proceedings of the AAAI Conference on Artificial Intelligence. Vol. 37. No. 8. 2023.

[F] An, Yongqi, et al. "Fluctuation-based adaptive structured pruning for large language models." Proceedings of the AAAI Conference on Artificial Intelligence. Vol. 38. No. 10. 2024.

[G] Gao, Shangqian, et al. "Structural alignment for network pruning through partial regularization." Proceedings of the IEEE/CVF International Conference on Computer Vision. 2023.

---

### Official Review · Reviewer_G2u2 · 2024-11-04

**Soundness:** 2
**Presentation:** 2
**Contribution:** 2
**Rating:** 3
**Confidence:** 4

**Summary:**

The authors introduce neuron entropy as a quantitative measure of network expressiveness. Observing that higher neuron entropy correlates with better generalization, they propose HENP, a method that dynamically prunes inactive ("dying") neurons and sparsifies the network during training.

**Strengths:**

The proposed HENP-1 and HENP-2 approaches outperform other structured pruning methods across a range of sparsity levels, demonstrating effectiveness and robustness.

**Weaknesses:**

* The transition between Sections 3 and 4 is unclear. Section 3, on its own, lacks relevance for understanding the following sections. A more comprehensive explanation of the problem formulation would enhance readability.
* Only one architecture and one dataset are considered, limiting the generalizability of the method.
* Figure 2 is based on a single trial, which does not rule out the possibility of randomness influencing the results.
* In general, the figures lack clarity and conciseness, with redundant information. For instance, Figure 3 is difficult to interpret due to the presence of too many overlapping lines.

**Questions:**

* Based on Lines 164-165, could the authors provide a more rigorous proof showing why a higher capacity for capturing complex zones correlates with improved model performance?
* In Lines 214–215, could the authors clarify how the 50% is calculated?
* In Lines 244–245, what type of stability is being referred to? Does greater stability imply lower neuron entropy variance, or does it mean neuron entropy remains constant throughout training?
* In Lines 253–254, doesn’t the best-performing model exhibit the highest variance, which appears contrary to the conclusion here?
* For Figure 2, the findings would be more convincing if multiple trials were conducted to confirm a consistent pattern.
* In the result section, only ReLU and Leaky ReLU activations are considered in the paper. How would the method perform with activations that have more complex regions?

---

### Official Review · Reviewer_J9Dt · 2024-11-04

**Soundness:** 2
**Presentation:** 1
**Contribution:** 2
**Rating:** 3
**Confidence:** 3

**Summary:**

This paper proposes a framework for analyzing neural networks through the concept of dynamic and static neurons, focusing on neurons' activation patterns under different inputs. The authors propose the neuron entropy as a metric to quantify network expressiveness, which describes the uncertainty of the neuron activation zone. They also provide some theoretical analysis of the relationship between neuron entropy and model expressivity. A HENP (Hybrid Entropy-Norm Pruning) algorithm is proposed that combines neuron entropy with norms to eliminate unnecessary parameters from neural networks.

**Strengths:**

- Offers new insights into neural network behavior and expressiveness.
- Not only defines neuron entropy but also demonstrates its utility in evaluating network expressivity.
- Clear progression from theoretical foundations to practical applications.

**Weaknesses:**

1. Some descriptions in Section 4.2.1 regarding the relationship between neuron entropy and model performance appear inconsistent with the presented figures, which need clarification or correction.
For example, "The most pronounced differences between models are observed in the deeper layers (layer 11, Figure 2c), where higher neuron entropy correlates with improved accuracy on both training and testing datasets." However, in Figure 2c, the model with the best performance (batch size 64, red line) has the lowest neuron entropy. Likewise, the description “Moreover, when examining the variance of neuron entropy, we observe that the model with the best performance consistently exhibits the lowest variance across different layers.” is not shown in the Figure. How do you reconcile the apparent contradiction between your claims and the results shown in Figure 2? Could you explain whether this contradiction challenges your theory or if there are other factors at play?

2. Use of "pruning frequency" instead of "pruning interval" in Section 5.3 is technically incorrect. Going from pruning every epoch to pruning every 15 epochs cannot be described as "increasing the pruning frequency". I suggest revising the terminology throughout Section 5.3 and any related sections to ensure consistency and accuracy in describing the pruning schedule.

3. Experiments mainly focus on CIFAR-10. Could include more datasets.

4. Limited analysis of the computational overhead of calculating neuron entropy during training. Is there a trade-off between performance improvement and computational cost? Could you provide a quantitative analysis of the computational overhead introduced by neuron entropy calculation? How does this overhead scale with model size and batch size?
include a comparison of training times with and without this calculation, or a discussion of how the overhead scales with model size

5. While the appendix provides detailed theoretical foundations, some practical implementation details are still unclear. For example, how are the weights $w_c$ in the weighted entropy calculation determined in practice? Or could you include a brief discussion of any challenges or considerations in determining these weights?

6. Comparisons mainly focus on a few structured pruning methods. Could you add some comparisons with recent state-of-the-art pruning approaches, or at least add some description.

**Questions:**

Please see the weaknesses part.

---

### Note · Authors · 2024-11-23

I have read and agree with the venue's withdrawal policy on behalf of myself and my co-authors.